# Prevalence of aspiration pneumonia among stroke patients in Ethiopia: A systematic review and meta-analysis

Assefa Andargie Kassa[1]*, Getahun Gebre Bogale[2], Mekuanint Taddele[1], Tilahun Degu Tsega[1], Abebaw Molla[1], Wolde Melese[1], Segenet Zewdie[3]

1 Department of Public Health, College of Medicine and Health Science, Injibara University, Injibara, Ethiopia, 2 Department of Health Informatics, School of Public Health, Asrat Woldeyes Health Sciences Campus, Debre Berhan University, Debre Birhan, Ethiopia, 3 Department of Pharmacy, College of Medicine and Health Science, Injibara University, Injibara, Ethiopia

* assefaand@gmail.com

## Abstract

Aspiration pneumonia (AP) is a serious complication among stroke patients, increasing the risk of poor outcomes and death. Although a previous review in Ethiopia attempted to estimate its burden, it had methodological limitations. This study aimed to provide an updated and comprehensive estimate of the pooled prevalence of AP among stroke patients in Ethiopia. A systematic review and meta-analysis were conducted following PRISMA guidelines. Relevant studies were identified from electronic databases and grey literature. Data were analyzed using STATA version 17 with the *metaprop* package. A random-effects model with Freeman–Tukey double arcsine transformation was applied. Heterogeneity was assessed using I² and Cochran's Q test. Subgroup analysis, meta-regression, and leave-one-out sensitivity analysis were conducted. Publication bias was evaluated using funnel plot symmetry and Egger's test. A total of 27 studies involving 7,120 stroke patients were included. The pooled prevalence of AP was 24.26% (95% CI: 20.76%–27.95%), with significant heterogeneity (I²=91.96%; p<0.001). Subgroup analysis showed significant regional variation, with the highest prevalence in Harari (39.48%) and the lowest in Addis Ababa (16.31%). Meta-regression showed no significant associations with study year, sample size, sex proportion, ischemic stroke proportion, or mean age. Sensitivity analysis confirmed the robustness of the findings. No publication bias was detected. Aspiration pneumonia affects nearly one in four stroke patients in Ethiopia, with notable regional variation, highlighting the need for targeted prevention strategies.

**Registration (PROSPERO):** CRD42024498777

**Data availability statement:** The dataset is uploaded with the manuscript as supporting information.

**Funding:** The author(s) received no specific funding for this work.

**Competing interests:** The authors have declared that no competing interests exist.

## Introduction

Stroke is a major global health concern, contributing significantly to morbidity and mortality rates worldwide. In 2019, it accounted for 6.2 million deaths and 139.4 million disability-adjusted life years (DALYs), making it the second most common cause of death and the third most common cause of DALYs [1]. The total value of lost welfare due to stroke worldwide in 2019 was $2059.67 billion, equivalent to 1.66% of the global gross domestic product (GDP) [2]. Based on a review study on incidence and prevalence of stroke in Africa, the unstandardized annual incidence rate of stroke ranges from 25–260 per 100,000 from 1973 to 2013 [3]. On the other hand, a systematic review and meta-analysis published in 2023 reported a crude incidence rate of 106.5 per 100,000 persons [4]. Another study reported that the in-hospital mortality from stroke was 22% in Sub-Saharan Africa [5].

In Ethiopia, the burden of stroke is on the rise, with an increasing number of individuals affected by this debilitating condition [6]. Reports on in-hospital mortality from stroke slightly vary in Ethiopia. A systematic review and meta-analysis reported 14.03% [7], and another reported 18% in-hospital stroke mortality in Ethiopia [8].

In Ethiopia, the management of acute stroke is guided by the National Guideline on Major Non-Communicable Diseases, developed by the Federal Ministry of Health in 2016 [9]. This guideline provides standardized recommendations for the diagnosis and treatment of stroke, with an emphasis on early recognition, clinical stabilization, imaging-based classification, and appropriate pharmacological interventions. However, implementation remains challenged by limited diagnostic infrastructure, delayed patient presentation, and inadequate access to thrombolytic therapy and rehabilitation services. Strengthening stroke care pathways and health system capacity is essential to improve clinical outcomes in the Ethiopian context.

Despite advancements in stroke management, stroke patients often face a multitude of complications post-stroke, which can profoundly impact their quality of life and increase healthcare burden. Aspiration pneumonia (AP), increased intracranial pressure, seizure, deep vein thrombosis, hospital acquired infections and electrolyte imbalance are among the common complications experienced by stroke patients [3,10].

AP which is the most serious type of complication, refers to the microaspiration of bacterially rich gastrointestinal or oropharyngeal secretions into the lungs in amounts high enough to cause inflammation of the alveoli and systemic circulation [11–13]. Impaired swallowing (dysphagia), gastrointestinal disorders, dentition, impaired consciousness, impaired cough reflex, and age are common risk factors associated with aspiration pneumonia [11,13–15]. Patients who have had a stroke are more likely to experience dysphagia and impaired consciousness as a result of the neurological damage caused by the event. Dysphagic stroke patients run the danger of aspirating stomach contents. AP is more prevalent among dysphagic individuals [16–19]. Moreover, stroke induced immunosuppression; driven by sympathetic nervous system activation, impaired immune cell function, elevated cortisol levels, and increased cell apoptosis; heightens the risk of developing AP after stroke [20–24].

AP poses significant burdens for stroke patients, impacting their clinical outcomes and overall quality of life. The incidence of aspiration pneumonia in stroke patients ranges from 3.9% to 12%, contributing to increased morbidity and mortality rates [24,25]. Hospitalization from both community-acquired and hospital-acquired pneumonia is frequently caused by aspiration pneumonia [26,27]. Evidences show that post stroke pneumonia is mostly associated with aspiration [28] where 3% to 50% patients with stroke may develop AP [13].

A previous systematic review on aspiration pneumonia (AP) among stroke patients in Ethiopia has been published [29]; however, it is limited by major methodological flaws. It included studies with nonspecific outcomes (e.g., pneumonia and stroke-associated pneumonia) [30,31], used data from general medical admissions instead of stroke-specific populations [32], and incorporated multiple publications from similar datasets [33,34]. Several prevalence estimates were inaccurately extracted, and at least five eligible studies were missed. Our review addresses these issues through rigorous methodology and includes six additional recent studies, offering a more accurate and comprehensive synthesis of the evidence on AP among stroke patients in Ethiopia.

This review aims to estimate the prevalence of AP among stroke patients in Ethiopia by integrating the body of available knowledge. Studying aspiration pneumonia among stroke patients is crucial for reducing its incidence, improving patient outcomes, and optimizing healthcare resources. It leads to better preventive strategies, enhanced rehabilitation techniques, and overall improved quality of life for stroke patients.

## Methods

### Protocol and registration

This systematic review and meta-analysis adhered to the Preferred Reporting Items for Systematic Reviews and Meta-Analyses (PRISMA) guidelines [35] (S1 Checklist, S2 Checklist). To ensure transparency and pre-defined methodological rigor, the study protocol was prospectively registered in the International Prospective Register of Systematic Reviews (PROSPERO) under registration number CRD42024498777.

### Eligibility criteria

Inclusion criteria were defined using the Condition, Context and Population (CoCoPop) framework [36].

- **Condition**: The primary condition is aspiration pneumonia (AP). We included all studies reporting AP as a primary outcome or a complication among individuals diagnosed with stroke. Aspiration pneumonia was defined based on clinical, radiological, or diagnostic criteria as reported by the original studies.

- **Context**: Studies conducted within the geographical boundaries of Ethiopia, regardless of healthcare setting.

- **Population**: Studies involving adult patients (aged ≥18 years) who had been clinically diagnosed with stroke, including ischemic stroke, hemorrhagic stroke, or unspecified types. Studies were eligible regardless of the sex of participants, presence of comorbidities, or stroke severity.

- **Study Types**: We included observational studies that reported the prevalence of aspiration pneumonia, including: Cross-sectional studies, Prospective or retrospective cohort studies and Case-control studies.

- **Outcome Measure**: Studies must have reported prevalence data on aspiration pneumonia among stroke patients, either as a primary outcome or as extractable from the study results.

- **Time Frame**: No restriction was applied on the year of publication. Studies published until May 10, 2025 were included.

- **Language:** We did not apply any language restriction during the literature search. However, all eligible studies identified and included in the final review were published in English.

Whereas, studies conducted among patients with a specific condition, studies focusing on pediatric populations or non-stroke-related causes of pneumonia, case reports, reviews, editorials, or commentaries that do not provide original data and studies that do not separate data on aspiration pneumonia from other types of pneumonia in stroke patients were excluded. In cases of multiple publications using the same dataset, only the most complete and recent version was included.

## Search strategy

A comprehensive search strategy was developed to identify relevant studies. Databases such as PubMed, Scopus, Cumulative Index to Nursing and Allied Health Literature (CINAHL), Semantic scholar, crossref, and Google scholar were systematically searched. We used the Harzing's Publish or Perish (Windows GUI Edition) Version 8.2.3944.8118 software to search Scopus, Semantic Scholar, crossref and Google Scholar databases. The search strategy included Medical Subject Headings (MeSH) terms and keywords related to stroke, aspiration pneumonia, and Ethiopia. The search strings were developed using the 'AND' and 'OR' Boolean operators. Further studies were sought using a snowballing process from references of primary studies. The search was conducted from inception to May 10, 2025 for all databases, with no language or publication status restrictions applied. Filters were applied for human species and adult (18 + years). The search strategy for PubMed is presented below and others are presented in a supplementary file (S1 Text).

((stroke[mesh] OR stroke[ti] OR "post-stroke") AND (complication OR sequelae OR comorbidity OR Outcome* OR Pneumonia OR "aspiration pneumonia")) AND (Ethiopia) Filters: Humans, Adult: 18 + years

## Study selection

Search results from databases were exported to EndNote 20.5 (Clarivate LLC, 2022, US) and duplicates were identified. Two independent reviewers (AAK and SZ) conducted the screening process. Titles and abstracts were initially screened for eligibility, followed by a full-text assessment of potential studies. Inclusion decisions were based on predefined criteria, with disagreements resolved through consensus.

## Quality assessment

The methodological quality of included studies was assessed using the Joanna Briggs Institute (JBI) quality assessment tool for studies reporting prevalence data [36]. This tool evaluates the methodological rigor of observational studies based on criteria related to sample representativeness, study design, data collection methods, and statistical analysis in nine items. Each of the nine items in the checklist was scored as "Yes" (1 point), "Unclear" (0 points), or "No" (–1 point). A total score was calculated for each study and converted into a percentage. Based on the percentage score, studies were classified as high quality (≥ 80%), Moderate quality (60–79%) and Low quality (< 60%) (S2 Text). The quality assessment was done by two authors (TDT and WM) and discordances were resolved through discussion.

## Data extraction

Data extraction was independently performed by two reviewers (AAK and GGB) using a format on Excel spreadsheet with any discrepancies resolved through discussion. The extraction format included study characteristics (author, year of publication/posting, publication status, journal/data base, study setting, study duration, study design, data source, primary outcome and sample size), participant characteristics (mean age, proportion of male and female, proportion of urban and rural, proportion of ischemic vs hemorrhagic stroke) and study outcome (effect size) aspiration pneumonia prevalence (S2 Table). Missing data in the included studies were assessed, and where possible, study authors were contacted to obtain additional information. In cases where the mean age was not provided, we followed the recommendation of Hozo et al. and converted to the mean age [37]. In the meta-analysis, only studies with complete outcome data were included.

### Data synthesis and analysis

A narrative synthesis of included studies was conducted to summarize the findings related to post-stroke aspiration pneumonia among stroke patients in Ethiopia. The meta-analysis was performed using STATA 17 (StataCorp LLC, College Station, Texas 77845 USA). The *metaprop* package was employed to pool prevalence estimates and calculate corresponding 95% confidence intervals, accounting for between-study heterogeneity using a random-effects model. Quantitative data synthesis utilized a random-effects meta-analysis model with the Restricted Maximum Likelihood (REML) between studies variance estimation technique, considering potential heterogeneity among studies. The Freeman–Tukey transformation was employed as a variance-stabilizing transformation. This transformation was preferred for its ability to address both the problem of confidence limits outside the [0,100] range and that of variance instability [38]. To facilitate interpretation, we reported the back transformed proportions in percentage. Heterogeneity was assessed using the $I^2$ statistic and test of homogeneity of effect sizes was done using the Cochran's homogeneity test (Q statistic) at the 0.05 significance level. Subgroup analyses were conducted based on region, study design, data source and publication status. A random effects meta-regression was performed using sample size, year of publication, proportion of male patients, proportion of ischemic stroke, and mean age. Tables, figures and forest plots were used to display the results.

### Publication bias

Publication bias was assessed using funnel plots and asymmetry was tested using Egger's regression test with a 0.05 significance level [39].

### Sensitivity analysis

Sensitivity analyses were conducted to explore the impact of methodological quality and study design on overall results. We performed the leave-one-out analysis to check the influence of individual studies on the pooled effect size.

### Ethical considerations

Since the study analyzed data that was made available to the public, ethical approval was not considered necessary.

## Results

### Study selection

A total of 679 records identified from database searches. After removing 110 duplicate records, we screened 569 records for titles and abstracts (S1 Table). We sought retrieval of full texts for 60 records and each one were assessed for eligibility. Finally, the review included 27 studies. We looked through all of the papers that cited the studies that were first included, as well as the references of the included studies. However, these tracking efforts did not turn up any more article that meet the inclusion criterion (Fig 1).

### Study characteristics

A total of 27 studies conducted between 2015 and 2025 were included in this systematic review and meta-analysis. The studies were geographically diverse, covering multiple regions of Ethiopia: Amhara (n = 9) [40–48], Oromia (n = 9) [49–57], Tigray (n = 3) [34,58,59], Harari (n = 3) [60–62], Addis Ababa (AA) (n = 2) [63,64], and Sidama (n = 1) [65]. Regarding study design, retrospective cross-sectional (RCS) [34,40,43,48,50,52,55,60,61,63] and retrospective cohort (RC) [42,44–47,53,59,62] studies were the most commonly employed designs, followed by prospective cohort (PC) [51,54,58,64], cross-sectional (CS) [41,49,57,65] and case-control [56] studies. The majority of studies utilized secondary data sources (patient records) [34,40,42–46,48–50,52–56,59–63,65], while a few incorporated primary and institutional records [41,47,51,57,58,64].

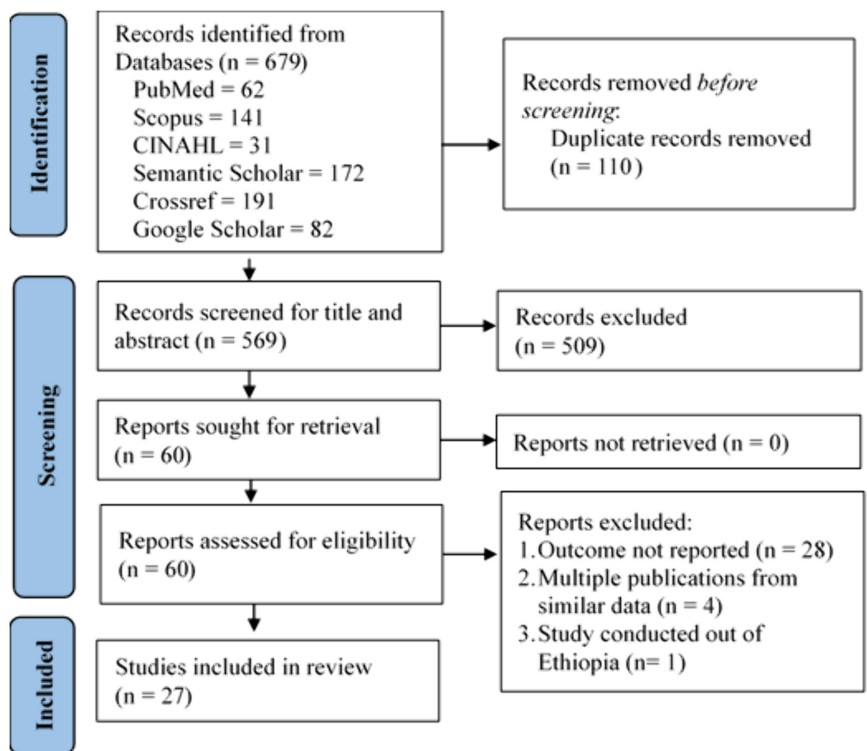

**Fig 1. PRISMA flow diagram used to select studies for the systematic review and meta-analysis of the prevalence of aspiration pneumonia among stroke patients in Ethiopia.**

The combined sample size across all included studies was 7,120 patients. Sample sizes ranged from 71 [64] to 597 [43]. The primary outcomes assessed were diverse, including in-hospital mortality, stroke complications (e.g., aspiration pneumonia, renal dysfunction), stroke management and outcomes, and clinical characteristics such as risk factors and stroke type distribution. Only two studies reported aspiration pneumonia as their primary outcome [44,48] (Table 1).

### Participant characteristics

The proportion of male participants ranged from 41.7% [34] to 72.5% [50], with a pooled mean of approximately 56.1%. Urban residency varied widely ranging from 22.9% [44] to 60.1% [54], reflecting demographic heterogeneity.

The mean age of stroke patients ranged between 52.5 years [63] and 68 years [42], with the majority of studies reporting mean ages in the early to mid-60s, suggesting a relatively younger stroke population compared to global estimates.

Ischemic stroke (IS) was generally more common than hemorrhagic stroke (HS). The proportion of IS ranged from 42.9% [65] to 100% [46], while HS ranged from 0% to 57.1%. The average IS and HS proportions across studies were approximately 61.46% and 36.4%, respectively. The prevalence of aspiration pneumonia (AP) among stroke patients was reported in all included studies, with a wide range between 7.27% [55] and 42.78% [62], indicating considerable variation across settings and patient populations (Table 2).

### Quality assessment of included studies

Of the 27 included studies, 11 were rated as high quality [34,43,44,48,51–53,56,59,62,65], 10 as moderate quality [40–42,45–47,55,57,58,61], and 6 as low quality [49,50,54,60,63,64]. Most high-quality studies provided clear definitions

**Table 1. The characteristics of studies included in the systematic review and meta-analysis on the prevalence of aspiration pneumonia among stroke patients in Ethiopia.**

| Author (year) | Publication Status | Region | Study Design | Data source | Sample size | Primary outcome studied | Quality (%) | Rating |
|---|---|---|---|---|---|---|---|---|
| Abdella (2019) | Pre-Print | Amhara | RCS | PR | 151 | In-hospital death | 77.78 | Moderate |
| Adem (2023) | Published | Harari | RCS | PR | 112 | In-hospital death | 55.56 | Low |
| Asgedom (2020) | Published | Tigray | RCS | PR | 216 | Medical complications, death | 88.89 | High |
| Asres (2020) | Published | AA | RCS | PR | 170 | Stroke prevalence, nursing management and outcomes | 55.56 | Low |
| Ayehu (2022) | Published | Amhara | CS | PRI | 554 | Risk profile, and clinical presentation | 66.67 | Moderate |
| Ayele (2023) | Published | Harari | RCS | PR | 290 | Poor treatment outcome | 66.67 | Moderate |
| Bekele (2023) | Published | Oromia | CS | PR | 135 | Stroke treatment outcome | 55.56 | Low |
| Beyene (2021) | Published | Oromia | RCS | PR | 153 | Stroke treatment outcome | 55.56 | Low |
| Fekadu (2019) | Published | Oromia | PC | PRI | 116 | Stroke management and complications | 88.89 | High |
| Gadisa (2020) | Published | Oromia | RCS | PR | 111 | Stroke treatment outcome | 88.89 | High |
| Gidey (2023) | Published | Tigray | PC | PRI | 272 | Stroke characteristics, risk factors and mortality | 66.67 | Moderate |
| Greffie (2015) | Published | Amhara | RC | PR | 98 | Clinical pattern, risk factors and outcomes of stroke | 66.67 | Moderate |
| Gufue (2020) | Published | Tigray | RC | PR | 503 | Stroke survival | 88.89 | High |
| Kefale (2020) | Published | Amhara | RCS | PR | 597 | Stroke treatment outcomes | 88.89 | High |
| Lidetu (2023) | Published | Amhara | RC | PR | 568 | Incidence of aspiration pneumonia | 100.00 | High |
| Mamushet (2015) | Published | AA | PC | PRI | 71 | Medical and neurological complications | 55.56 | Low |
| Mosisa (2023) | Published | Oromia | RC | PR | 480 | Stroke mortality | 88.89 | High |
| Mulugeta (2020) | Published | Amhara | RC | PR | 162 | Magnitude, risk factors and outcomes of stroke | 66.67 | Moderate |
| Teshome (2023) | Thesis | Sidama | CS | PR | 266 | Treatment outcome of stroke | 88.89 | High |
| Wubshet (2023) | Published | Oromia | PC | PR | 153 | Stroke clinical characteristics and short term outcomes | 55.56 | Low |
| Zewudie (2020) | Published | Oromia | RCS | PR | 220 | Stroke treatment outcome | 77.78 | Moderate |
| Abas (2024) | Published | Harari | RC | PR | 395 | Mortality | 88.89 | High |
| Addisu (2025) | Published | Amhara | RC | PR | 278 | Length of stay | 66.67 | Moderate |
| Ayehu (2025) | Published | Amhara | RC | PRI | 403 | Poor sleep quality | 66.67 | Moderate |
| Nigus (2024) | Published | Amhara | RCS | PR | 242 | Post stroke Aspiration Pneumonia | 100.00 | High |
| Nigussie (2024) | Published | Oromia | CS | PRI | 200 | Renal dysfunction and in-hospital mortality | 66.67 | Moderate |
| Hussein (2024) | Preprint | Oromia | CaC | PR | 204 | In-hospital mortality | 88.89 | High |

Abbreviations:

AA: Addis Ababa; CS: Cross sectional; PC: Prospective cohort; PR: Patient record; PRI: Patient record and interview; RC: Retrospective cohort; RCS: Retrospective cross sectional;

of the target population, employed appropriate sampling and statistical methods. Common limitations in lower-quality studies included insufficient reporting on sample representativeness and strategies to address response bias. The overall quality of the evidence was acceptable, and no study was excluded based on quality. However, study quality was explored further in sensitivity analyses (Table 1, S2 Text).

## Prevalence of aspiration pneumonia among stroke patients

The pooled prevalence of AP among stroke patients was estimated to be 24.26% (95% CI: 20.76%, 27.95%) using a random effects model. A Freeman–Tukey double arcsine transformation was applied to stabilize the variance of proportions. The analysis revealed significant between-study heterogeneity ($I^2 = 91.96\%$; $p < 0.001$), indicating substantial variability in prevalence estimates across the included studies (Fig 2).

**Table 2. Study participant characteristics that were part of the meta-analysis and systematic review on the prevalence of aspiration pneumonia in Ethiopian stroke patients.**

| Author (year) | Male % | Urban % | Mean age | IS % | HS % | AP (Count) | AP % |
|---|---|---|---|---|---|---|---|
| Abdella (2019) | 49.70 | 35.10 | 65.00 | 60.30 | 39.70 | 45 | 29.80 |
| Adem (2023) | 61.60 | NR | 60.32 | 43.75 | 56.25 | 39 | 34.82 |
| Asgedom (2020) | 41.70 | 46.80 | 61.20 | 55.60 | 44.40 | 92 | 42.59 |
| Asres (2020) | 57.10 | NR | 52.49 | 51.20 | 37.60 | 18 | 10.59 |
| Ayehu (2022) | 46.75 | 31.60 | 61.00 | 60.29 | 39.71 | 160 | 28.88 |
| Ayele (2023) | 62.80 | 29.00 | 54.70 | 58.60 | 41.40 | 110 | 37.93 |
| Bekele (2023) | 63.00 | 37.80 | 57.90 | 64.40 | 25.20 | 17 | 12.59 |
| Beyene (2021) | 72.50 | 42.00 | 57.30 | 73.20 | 26.80 | 46 | 30.07 |
| Fekadu (2019) | 62.90 | 27.60 | 55.14 | 51.70 | 48.30 | 23 | 19.83 |
| Gadisa (2020) | 49.50 | 36.00 | 63.40 | 80.20 | 18.00 | 24 | 21.62 |
| Gidey (2023) | 42.60 | 53.70 | 64.30 | 62.90 | 37.10 | 32 | 11.76 |
| Greffie (2015) | 46.90 | 55.40 | 68.00 | 69.40 | 30.60 | 19 | 19.39 |
| Gufue (2020) | 50.10 | 56.70 | 64.30 | 56.60 | 43.40 | 183 | 36.38 |
| Kefale (2020) | 53.10 | 46.60 | 61.08 | 65.70 | 31.50 | 125 | 20.94 |
| Lidetu (2023) | 60.21 | 22.90 | 67.00 | 69.01 | 30.99 | 131 | 23.06 |
| Mamushet (2015) | 60.60 | NR | 52.76 | 60.50 | 39.40 | 24 | 33.80 |
| Mosisa (2023) | 62.29 | 40.63 | 55.43 | 54.58 | 45.42 | 127 | 26.46 |
| Mulugeta (2020) | 46.30 | 46.30 | 58.30 | 50.00 | 30.00 | 32 | 19.75 |
| Teshome (2023) | 58.60 | 40.60 | 56.14 | 42.90 | 57.10 | 47 | 17.67 |
| Wubshet (2023) | 52.90 | 60.10 | 57.00 | 48.40 | 51.60 | 32 | 20.92 |
| Zewudie (2020) | 67.30 | 35.10 | 62.33 | 70.90 | 29.10 | 16 | 7.27 |
| Abas (2024) | 62.50 | 51.10 | 61.02 | 64.60 | 35.40 | 169 | 42.78 |
| Addisu (2025) | 59.70 | 34.50 | 66.07 | 100.00 | 0.00 | 57 | 20.50 |
| Ayehu (2025) | 44.00 | 32.27 | 61.30 | 67.70 | 32.30 | 82 | 20.35 |
| Nigus (2024) | 55.80 | 42.10 | 61.15 | 60.30 | 39.70 | 56 | 23.14 |
| Nigussie (2024) | 64.50 | 34.00 | 60.50 | 58.00 | 42.00 | 71 | 35.50 |

Abbreviations

AP: Aspiration pneumonia; HS: Hemorrhagic stroke; IS: Ischemic stroke; NR: Not reported

## Subgroup analysis

Subgroup analysis was performed to explore the source of heterogeneity by data source, study design, publication status, quality rating and region. There was no statistically significant difference in effect size across groups for data source, study design, publication status or quality rating.. Subgroup analysis by region revealed a statistically significant variation in the prevalence of AP among stroke patients. The highest pooled prevalence was observed in Harari (39.48%; 95% CI: 35.17%–43.87%), followed by Tigray (29.09%; 95% CI: 12.64%–49.01%), Amhara (22.9%; 95% CI: 20.49%–25.39%), Oromia (20.95%; 95% CI: 15.81%–26.60%), and Addis Ababa (16.31%; 95% CI: 11.85%–21.30%). The degree of heterogeneity varied across regions, with moderate heterogeneity in Amhara ($I^2 = 58.63\%$) and high heterogeneity in Oromia ($I^2 = 88.7\%$). Heterogeneity estimates for Harari, Tigray, and Addis Ababa could not be determined due to the limited number of studies included in each subgroup (Fig 3).

To further explore sources of heterogeneity, a meta-regression analysis was conducted using continuous study-level covariates, including study year, sample size, proportion of male participants, proportion of ischemic stroke cases, and mean age of participants. However, none of these variables showed a statistically significant association with the prevalence of aspiration pneumonia, indicating that they did not explain the observed between-study heterogeneity.

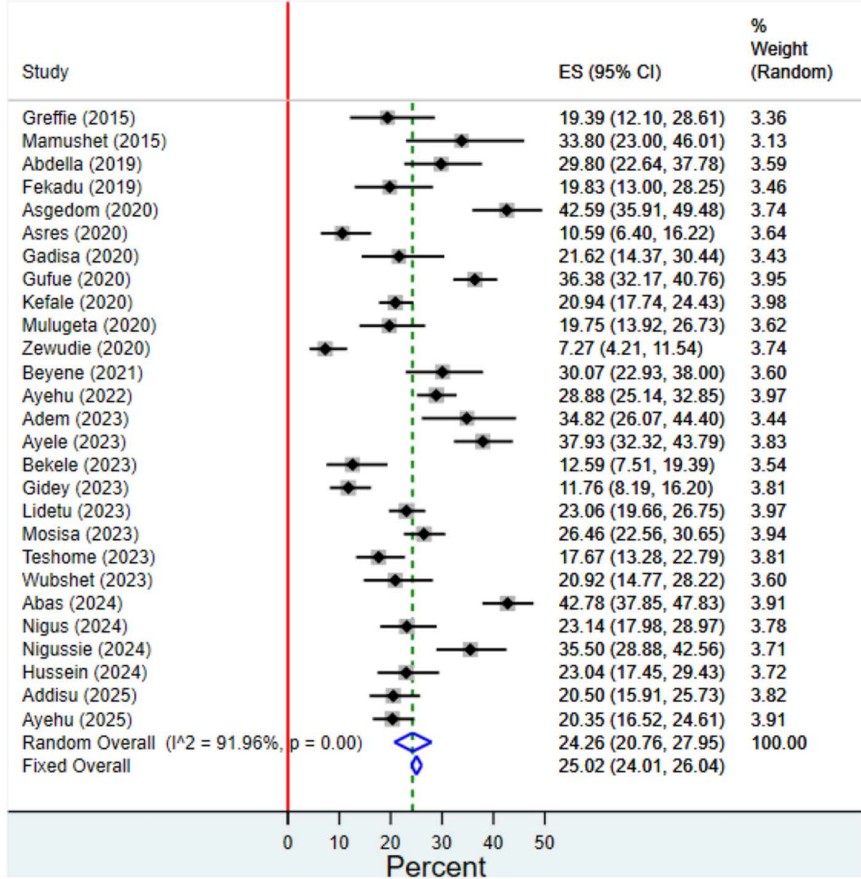

**Fig 2. Forest plot of the pooled prevalence of aspiration pneumonia among stroke patients in Ethiopia.**

## Bias assessment

Assessment of publication bias using a funnel plot showed a symmetrical distribution of studies, and Egger's regression test was not statistically significant (p = 0.454), indicating no evidence of publication bias (Fig 4).

## Sensitivity analysis

Sensitivity analysis was performed using the leave-one-out method to assess the robustness of the pooled prevalence estimate. The results showed that no single study significantly influenced the overall pooled prevalence of aspiration pneumonia, indicating the stability and reliability of the meta-analysis findings (Fig 5).

## Discussion

This systematic review and meta-analysis aimed to determine the pooled prevalence of aspiration pneumonia among stroke patients in Ethiopia. The analysis included 27 studies encompassing a total of 7,120 stroke patients, with reported prevalence of aspiration pneumonia ranging from 7.27% to 42.78%.

In this study, the pooled prevalence estimate of aspiration pneumonia (AP) among stroke patients in Ethiopia was 24.26% (95% CI: 20.76–27.95), which is lower than the 31.65% (95% CI: 25.30–38.01) reported in a previous systematic review [29]. Notably, the confidence intervals of the two estimates do not fully overlap, suggesting a statistically

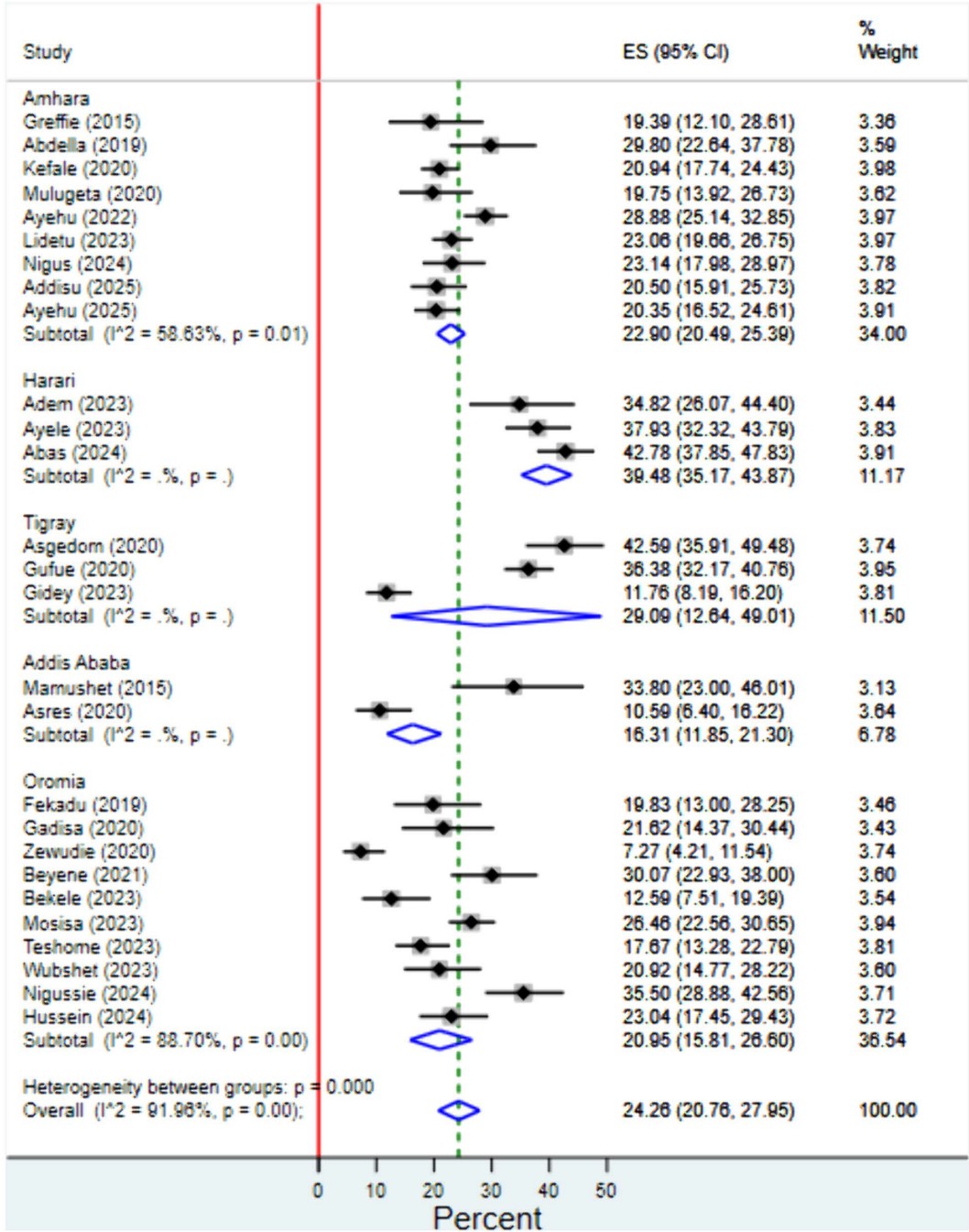

**Fig 3. Forest plot for the sub-group analysis of the prevalence of aspiration pneumonia among stroke patients by study region in Ethiopia.**

meaningful difference between the findings. This discrepancy may stem from key methodological differences. The earlier review included studies with broader or unclear pneumonia definitions, such as general or stroke-associated pneumonia rather than strictly aspiration pneumonia, potentially leading to overestimation. It also contained duplicate datasets and inaccurate data extraction, and missed several eligible studies available during its search period. Our review addressed

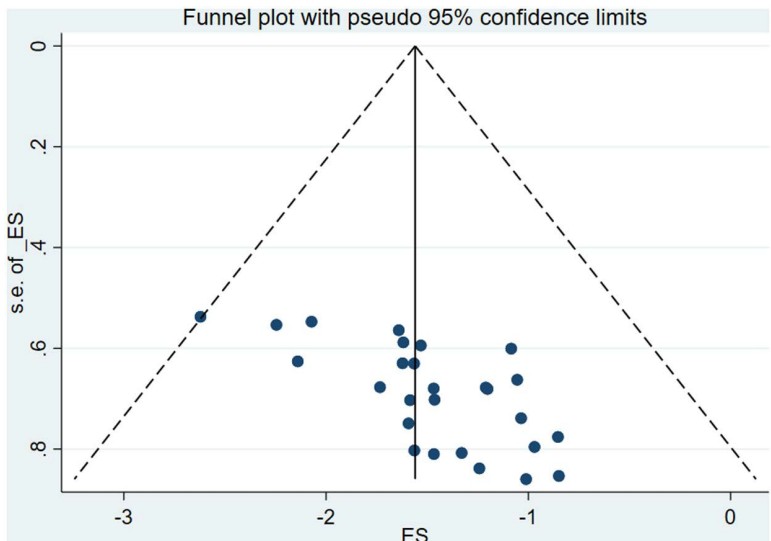

**Fig 4. Funnel plot of included studies in the systematic review and meta-analysis on the prevalence of aspiration pneumonia among stroke patients in Ethiopia.**

these limitations by applying stricter inclusion criteria focused solely on aspiration pneumonia, verifying data accuracy, and incorporating six additional recent studies. Consequently, our estimate likely reflects a more accurate and current burden of AP among stroke patients in Ethiopia.

This finding highlights the significant burden of AP in this population, which warrants urgent attention from healthcare providers and policymakers. The prevalence is concerning, as AP can lead to severe complications, prolonged hospital stays, increased healthcare costs, and higher mortality rates among stroke patients [66]. The finding underscores the importance of monitoring and managing this complication in stroke patients to improve outcomes and reduce mortality.

When compared to global and regional data, the prevalence of AP among stroke patients in Ethiopia is in line with a study in Japan [67]. However, it appears to be higher compared to studies conducted in other high-income countries, generally ranging from 5% to 15% [68,69]. This discrepancy may be attributed to differences in healthcare infrastructure, the availability of rehabilitation services, and early intervention strategies [70]. In many high-income settings, stroke units are well-equipped with multidisciplinary teams that provide comprehensive care, including the screening and management of dysphagia, which is a significant risk factor for AP [17].

Several factors may contribute to the high prevalence of AP among stroke patients in Ethiopia. These include limited access to specialized stroke care, inadequate screening and management of dysphagia, poor nutritional status, and suboptimal post-stroke rehabilitation services [70]. The Ethiopian healthcare system faces numerous challenges that may impact the management of stroke and its complications. These challenges include a shortage of trained healthcare professionals, limited resources, lack of standardized stroke management procedures, inadequate access to specialized care (particularly in rural areas) and insufficient infrastructure [71,72]. On the other hand, many stroke patients may not receive timely medical intervention, which increases the risk of complications including aspiration pneumonia due to delayed medical attention. High patient-to-staff ratios in Ethiopian hospitals, especially in public healthcare facilities, make it difficult for healthcare workers to provide the necessary attention and monitoring needed to prevent aspiration [73,74].

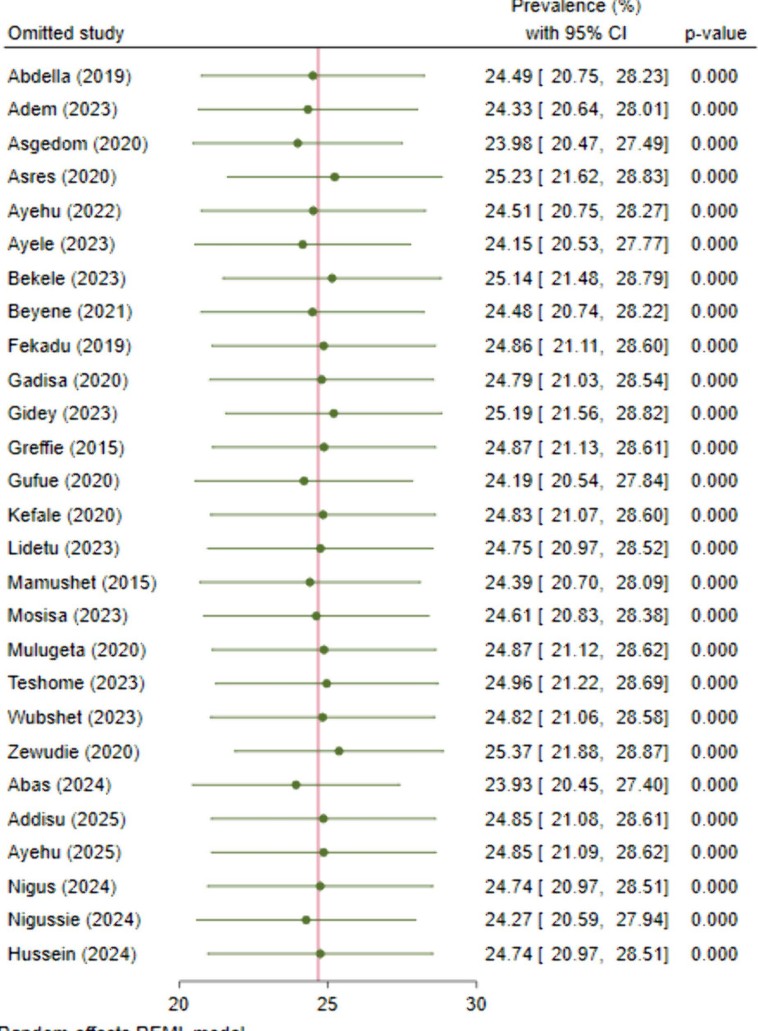

**Fig 5. Leave-one-out sensitivity analysis of prevalence of aspiration pneumonia among stroke patients in Ethiopia.**

Our subgroup analysis revealed significant regional variation in the prevalence of aspiration pneumonia among stroke patients in Ethiopia. The highest prevalence was observed in Harari (39.48%), followed by Tigray (29.09%), Amhara (22.9%), Oromia (20.95%), and Addis Ababa (16.31%). This variation suggests potential differences in healthcare infrastructure, diagnostic capacity, stroke management protocols, and post-stroke care practices across regions.

The notably high prevalence in Harari may reflect either a genuinely higher burden of AP or better clinical recognition and documentation compared to other regions. For instance, Hariri region has the lowest number of hospitals compared to other regions implying that most complicated patients may be referred to these small number of hospitals [75]. Conversely, the relatively lower prevalence in Addis Ababa might be attributed to better access to acute stroke care, including early dysphagia screening and aspiration prevention strategies in tertiary hospitals.

The moderate heterogeneity observed in regions like Amhara ($I^2 = 58.63\%$) and high heterogeneity in Oromia ($I^2 = 88.7\%$) also indicate variability within regions, possibly due to differences in study design, sample size, or healthcare

facility type (e.g., referral vs. primary hospitals). In Tigray, while the point estimate was high, the wide confidence interval suggests uncertainty, likely due to a smaller number of studies.

These findings underscore the importance of tailoring clinical and public health interventions based on local epidemiology. Regions with higher AP prevalence may benefit from strengthened capacity for early dysphagia screening, better stroke unit organization, and staff training. Future research should further explore region-specific risk factors, including clinical, demographic, and health system-related determinants, to guide more targeted and equitable stroke care strategies in Ethiopia.

## Limitations

This review has several limitations. First, most studies did not specify when aspiration pneumonia occurred, limiting interpretation of its timing relative to stroke onset. Second, there was no standardized definition of aspiration pneumonia across studies, contributing to outcome variability. Third, many studies were retrospective and reported aspiration pneumonia as a secondary outcome, which may affect data accuracy. Fourth, stroke phase and patient survival status were often unclear, making it difficult to assess population comparability. Lastly, limited data prevented subgroup analyses by stroke type, severity, or pneumonia onset, which may influence prevalence estimates.

## Conclusion

This systematic review and meta-analysis reveal a substantial burden of aspiration pneumonia among stroke patients in Ethiopia, with significant regional disparities. Addressing these disparities through targeted interventions, improved healthcare infrastructure, and increased public awareness is essential to reducing the incidence of AP and improving the overall health outcomes of stroke patients in Ethiopia. Future research should focus on identifying region-specific risk factors and developing tailored prevention and management strategies to mitigate this public health challenge.

## Supporting information

**S1 Text. The strategies for data bases searches.**
(DOCX)

**S2 Text. The quality assessment score of included studies.**
(DOCX)

**S1 Checklist. PRISMA 2020 Checklist.**
(DOCX)

**S2 Checklist. PRISMA 2020 for abstracts Checklist.**
(DOCX)

**S1 Data. Dataset.**
(XLS)

**S1 Table. List of studies identified in the literature search.**
(XLSX)

**S2 Table. Data extraction form.**
(XLSX)

## Author contributions

**Conceptualization: Assefa Andargie Kassa.**

**Data curation: Assefa Andargie Kassa.**

**Formal analysis:** Assefa Andargie Kassa, Tilahun Degu Tsega.

**Investigation:** Assefa Andargie Kassa.

**Methodology:** Assefa Andargie Kassa, Mekuanint Taddele, Segenet Zewdie.

**Project administration:** Assefa Andargie Kassa, Abebaw Molla, Segenet Zewdie.

**Software:** Assefa Andargie Kassa.

**Supervision:** Getahun Gebre Bogale.

**Validation:** Wolde Melese.

**Visualization:** Assefa Andargie Kassa, Segenet Zewdie.

**Writing – original draft:** Assefa Andargie Kassa.

**Writing – review & editing:** Getahun Gebre Bogale, Mekuanint Taddele, Tilahun Degu Tsega, Abebaw Molla, Wolde Melese, Segenet Zewdie.

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
