## [Decision Letter · Decision Letter 0]

PGPH-D-24-01656

Prevalence of aspiration pneumonia among stroke survivors in Ethiopia: A systematic review and meta-analysis

Dear Dr. Kassa,

Thank you for submitting your manuscript to PLOS Global Public Health. After careful consideration, we feel that it has merit but does not fully meet PLOS Global Public Health’s publication criteria as it currently stands. Therefore, we invite you to submit a revised version of the manuscript that addresses the points raised during the review process.

Please note that we have only been able to secure a single reviewer to assess your manuscript. We are issuing a decision on your manuscript at this point to prevent further delays in the evaluation of your manuscript. Please be aware that the editor who handles your revised manuscript might find it necessary to invite additional reviewers to assess this work once the revised manuscript is submitted. However, we will aim to proceed on the basis of this single review if possible. 

The reviewer has provided some comments which are available below. They have suggested including a more detailed discussion of the some of the disparities noted and limitations of the study. Please review their comments and make the appropriate revisions. 

We look forward to receiving your revised manuscript.

Kind regards,

Emma Campbell, Ph.D

Staff Editor

Journal Requirements:

1. As required by our policy on Data Availability, please ensure your manuscript or supplementary information includes the following: 

Reviewers' comments:

Reviewer's Responses to Questions

**Comments to the Author**

1. Does this manuscript meet PLOS Global Public Health’s publication criteria?

Reviewer #1: Yes

2. Has the statistical analysis been performed appropriately and rigorously?

Reviewer #1: Yes

3. Have the authors made all data underlying the findings in their manuscript fully available (please refer to the Data Availability Statement at the start of the manuscript PDF file)?

Reviewer #1: Yes

4. Is the manuscript presented in an intelligible fashion and written in standard English?

Reviewer #1: Yes

Reviewer #1: The work presented for review is interesting, addressing lesser-known non-neurological complications of stroke, such as aspiration pneumonia, which also impacts stroke treatment outcomes and mortality. The authors applied the PRISMA method, which is the appropriate approach for meta-analysis. Additionally, the choice of topic in this research is well-suited to the current needs in both medical and social contexts, particularly in African countries.

In the first part of the article, the authors thoroughly discuss the definition of stroke and its symptoms. Although this information is standard, it may be useful in a journal aimed at epidemiologists and global health professionals. The only suggestion for this section is to include visual impairments that occur during a stroke, such as homonymous hemianopia caused by brain damage beyond the optic chiasm, which significantly increases patient disability.

The epidemiological data presented also requires some commentary, as the authors report very high rates. Both the incidence of stroke and mortality rates significantly exceed the parameters observed in Europe, where the incidence is approximately 90-120 cases per 100,000 inhabitants per year, and mortality rates in most European countries are lower. This disparity also warrants discussion.

The methodology, developed according to the PRISMA system, is well-described, and the study period between June 2008 and September 2022 is sufficiently long to be representative of the methods and assumptions presented. It is evident from the supplementary data that the authors made significant efforts to adhere to PRISMA principles, perhaps even excessively so, which, while commendable, might not have been entirely necessary.

The results highlight the substantial variability in the occurrence of aspiration pneumonia, ranging from 7% to 43%, a range not typically seen in other studies. The authors rightly emphasize the significant heterogeneity of these results. Overall, the pooled proportion of 23% is notably high. One potential reason for such elevated rates, as suggested by the authors, is limited access to specialized stroke care, inadequate screening, and management of dysphagia. While this is a valid point, it also indicates a lack of standardized procedures in departments admitting stroke patients. For instance, a simple water-swallowing test conducted upon admission, which is neither time-consuming nor requires complex equipment, could effectively assess swallowing disorders and prevent the administration of oral substances, including medications and food.

It is positive that the authors noticed this problem and discussed it in some detail, although it is difficult to agree with the comment that a significant problem here is the lack of expensive and requiring specialist staff techniques such as fiberoptic endoscopic evaluation. In the early stages of diagnosing stroke patients, non-instrumental methods can also be quite effective in preventing aspiration pneumonia. I would expect the authors to provide some conclusions on this matter in their discussion and conclusions.

The authors' proposed further actions to determine the causes of the large discrepancies in aspiration pneumonia rates are purposeful and interesting from a public health perspective. It would be beneficial for the authors to hypothesize possible explanations for these differences. Although there may not be precise studies on this matter, the authors, given their familiarity with their healthcare system, should be able to offer some insights. For instance, could these differences be related to the average income per person in different regions, a higher level of healthcare, variations in insurance systems, or simply the introduction of specific medical procedures in some centers but not others?

In summary, the authors' initiative to explore this topic should be highly regarded. The work is interesting, methodologically sound, and, after addressing the minor issues raised in this and potentially other reviews, it certainly deserves consideration for publication

**Do you want your identity to be public for this peer review?** For information about this choice, including consent withdrawal, please see our Privacy Policy

Reviewer #1: **Yes: ** Prof. Radoslaw Kazmierski, MD, PhD, Department of Neurology, Collegium Medicum, University of Zielona Gora, Poland

---

## [Decision Letter · Decision Letter 1]

PGPH-D-24-01656R1

Prevalence of aspiration pneumonia among stroke survivors in Ethiopia: A systematic review and meta-analysis

Dear Dr. Kassa,

Thank you for submitting your manuscript to PLOS Global Public Health. After careful consideration, we feel that it has merit but does not fully meet PLOS Global Public Health’s publication criteria as it currently stands. Therefore, we invite you to submit a revised version of the manuscript that addresses the points raised during the review process.

Please provide a point by point response to the reviewer comments. 

We look forward to receiving your revised manuscript.

Kind regards,

Joanna Tindall, PhD

Staff Editor

Journal Requirements:

Additional Editor Comments (if provided):

Reviewers' comments:

Reviewer's Responses to Questions

**Comments to the Author**

Reviewer #2: (No Response)

publication criteria?

Reviewer #2: (No Response)

3. Has the statistical analysis been performed appropriately and rigorously?

Reviewer #2: (No Response)

4. Have the authors made all data underlying the findings in their manuscript fully available (please refer to the Data Availability Statement at the start of the manuscript PDF file)?

Reviewer #2: (No Response)

5. Is the manuscript presented in an intelligible fashion and written in standard English?

Reviewer #2: (No Response)

Reviewer #2: This is an interesting study on the incidence of aspiration pneumonia after stroke in the Ethiopian population. As I do not have the previous reviewer report, I do not know what previous comments have been made and whether they have been adressed now. So I cannot comment on that.

Major:

- the introduction section elaborates quite long on stroke, but should in my opinion speak more on the mechanisms of pneumonia after stroke. For example, immunosuppression after stroke is not mentioned, but is an important risk factor.

- how was aspiration pneumonia defined? What timeframe was used - first 7 days, admission, or other? What definition was used in the included studies? It is necessary for meta-analysis / pooling of data that similar definitions were used.

Many studies exists in other populations on 'stroke-associated pneumonia', with specific definition and a time limit of 7 days after stroke. See Stroke. 2015;46:2335-2340. DOI: 10.1161/STROKEAHA.115.009617

- table 2: as stroke severity is a major risk factor for pneumonia, this should be incorporated in the table. Also: medical history of obstructive lung disease, smoking yes/no, dysphagia, handicap before the stroke. In line 252-257 the authors describe subgroup analysis, most important predictors for pneumonia are missing (stroke severity, dysphagia, were studies performed on a general ward or on an ICU, were patient intubated/mechanically ventilated etc)

- why did the authors exclude studies on a single type of stroke? were studies solely on ischaemic stroke or haemorrhagic stroke excluded? Why?

- line 304-310: before drawing these conclusions it should be clear whether the included population is similar in both studies regarding stroke severity, dysphagia, medical history, ICU admission etc

minor

- discussion line 286: reference 61/62 report on pneumonia in general, not specific aspiration pneumonia. see also review westendorp et al, BMC Neurology 2015, for specific rates in general wards and on ICU's

- line 121-125 a different font size is used

-line 203 'after then'  next

- line 248-252 - description should be put in the methods section, not the results

**Do you want your identity to be public for this peer review?** For information about this choice, including consent withdrawal, please see our Privacy Policy

Reviewer #2: No

---

## [Decision Letter · Decision Letter 2]

PGPH-D-24-01656R2

Prevalence of aspiration pneumonia among stroke survivors in Ethiopia: A systematic review and meta-analysis

Dear Dr. Kassa,

Thank you for submitting your manuscript to PLOS Global Public Health. After careful consideration, we feel that it has merit but does not fully meet PLOS Global Public Health’s publication criteria as it currently stands. Therefore, we invite you to submit a revised version of the manuscript that addresses the points raised during the review process.

Please pay particular attention to Reviewer 2's comments, who has found that the previous issues were not sufficiently addressed. Some of these are key issues that must be addressed. Failure to do so could result in your manuscript being rejected in the next round. 

We look forward to receiving your revised manuscript.

Kind regards,

Daniel Parkes, PhD

Staff Editor

Journal Requirements:

Additional Editor Comments (if provided):

Reviewers' comments:

Reviewer's Responses to Questions

**Comments to the Author**

Reviewer #2: (No Response)

Reviewer #3: All comments have been addressed

publication criteria?

Reviewer #2: Partly

Reviewer #3: Yes

3. Has the statistical analysis been performed appropriately and rigorously?

Reviewer #2: I don't know

Reviewer #3: I don't know

4. Have the authors made all data underlying the findings in their manuscript fully available (please refer to the Data Availability Statement at the start of the manuscript PDF file)?

Reviewer #2: No

Reviewer #3: Yes

5. Is the manuscript presented in an intelligible fashion and written in standard English?

Reviewer #2: Yes

Reviewer #3: Yes

Reviewer #2: This is an article on an important topic that can give insight in the burden of pneumonia after stroke in Ethiopia. This would enable preventive measures and is insightful for policy makers.

The authors have addressed some of the previous comments and the article has improved by that, but there are still some major points that need to be addressed.

The line numbers refer to the line numbers in the track changes file (as no clean version is provided).

Major

- Major limitation for me is that it is not clear what population was studied. The authors describe ‘We included all studies 131 reporting AP as a complication among stroke patients admitted to hospitals’ but also name that only survivors of stroke were included. Also, no timeframe is given. For me it is unclear whether the occurrence of pneumonia during the first hospital stay for stroke was studied, or whether patients were also included with pneumonia months or even years after the stroke, for example in rehabilitation setting.

- Line 132 ‘was not considered necessary’. I disagree that it is not necessary. The incidence of pneumonia in stroke is the highest in the first week after stroke, when also the immune suppression is the most pronounced. Therefore it is likely that the incidence of aspiration pneumonia is different in the first week than in later course, for example weeks to months after the stroke. It is necessary to explain what population and what time phrame was used. If the duration of primary hospital admission was used, please add this to the text. If other, please explain.

- Line 143: what is meant by survivors of stroke? Survivor in what timeframe? How was this used for inclusion, for example, when a patient deceased in the first week because of aspiration pneumonia, was this patient included in the study or not? Could the authors explain how this was handled?

Minor

- As there is no clean version without the track changes, i cannot assess how the sentences are constructed in the clean version. In this version, there seem to be quite a lot of sentences that do not run well, for example the conclusion in the abstract, line 49 in the introduction section, line 225-227.

- Line 54 – is it ever expected?

- Line 53-54 duplicate the explanation in the lines thereafter (about the 2 types of stroke). Suggest to delete ‘a stroke … in the brain’ line 53-54.

- Line 59-68: this is a bit chaotic explanation and could be a lot shorter. For example ‘depending on the location, stroke can cause neurologic deficit such as aphasia, dysarthria, visual and motor or sensory disturbances, but also vertigo and headache.’

- Line 134-136 (‘stroke and … were included’) duplicates the sentences before, suggest to delete it.

- Line 140: suggest to update the search to look for new relevant articles.

- The authors should name that no definition for AP was used, but that any definition used in the included studies was sufficient, in the limitation section.

Reviewer #3: I appreciate the opportunity to review the manuscript titled "Prevalence of Aspiration Pneumonia Among Stroke Survivors in Ethiopia: A Systematic Review and Meta-Analysis." It is an interesting study that provides valuable insight into the incidence of stroke-associated pneumonia.

Here are my key observations:

The study effectively highlights the incidence of stroke-associated pneumonia, an important complication in stroke survivors.

The authors have adequately addressed the major concerns raised by previous reviewers.

The version I downloaded did not contain clear images, which may need to be addressed for clarity.

For a meta-analysis consisting of 21 studies, citing more than 75 references seems excessive. A more concise reference list may improve readability.

It would be beneficial to discuss standard care practices for stroke management and their correlation with the incidence of aspiration pneumonia. While the authors may not have access to all this data, mentioning standard care guidelines from local societies in the introduction would provide helpful context.

Overall, this study provides valuable contributions to the field. Please let me know if any further clarifications are needed.

**Do you want your identity to be public for this peer review?** For information about this choice, including consent withdrawal, please see our Privacy Policy

Reviewer #2: No

Reviewer #3: No

---

## [Decision Letter · Decision Letter 3]

Prevalence of aspiration pneumonia among stroke patients in Ethiopia: A systematic review and meta-analysis

PGPH-D-24-01656R3

Dear Mr. Kassa,

We are pleased to inform you that your manuscript 'Prevalence of aspiration pneumonia among stroke patients in Ethiopia: A systematic review and meta-analysis' has been provisionally accepted for publication in PLOS Global Public Health.

Best regards,

Julia Robinson

Executive Editor

Reviewer Comments (if any, and for reference):

Reviewer's Responses to Questions

**Comments to the Author**

Reviewer #3: All comments have been addressed

publication criteria?

Reviewer #3: Yes

3. Has the statistical analysis been performed appropriately and rigorously?

Reviewer #3: I don't know

4. Have the authors made all data underlying the findings in their manuscript fully available (please refer to the Data Availability Statement at the start of the manuscript PDF file)?

Reviewer #3: Yes

5. Is the manuscript presented in an intelligible fashion and written in standard English?

Reviewer #3: Yes

Reviewer #3: Thank you for the opportunity to review the article. I believe the authors have addressed all the comments made by prior reviewers. I have no concerns at this time

**Do you want your identity to be public for this peer review?** For information about this choice, including consent withdrawal, please see our Privacy Policy

Reviewer #3: No
